# The Efficacy of a Novel Water-Soluble Anti-Mycotoxin Solution in Improving Broiler Chicken Performance Under Mycotoxin Challenge

**DOI:** 10.3390/toxins17050212

**Published:** 2025-04-23

**Authors:** Sayantani Sihi Arora, Anusuya Debnath, Amrita Kumar Dhara, Sudipto Haldar, Raquel Codina Moreno, Insaf Riahi

**Affiliations:** 1Agrivet Research and Advisory Pvt Ltd., 714 Block A Lake Town, Kolkata 700089, India; amrita@agrivet.in (A.K.D.); sudipto@agrivet.in (S.H.); 2Department of Biotechnology, Brainware University, 398 Ramkrishnapur Road, Barasat, North 24 Parganas, Kolkata 700125, India; anusuyadebnath@yahoo.co.in; 3BIŌNTE Nutrition S.L., Mexico St. 33, 43204 Reus, Spain; raquel.codina@bionte.com (R.C.M.); insaf.riahi@bionte.com (I.R.)

**Keywords:** mycotoxin, liquid anti-mycotoxin solution, broiler chicken

## Abstract

Mycotoxins like aflatoxins (AFs), fumonisins (FBs), and ochratoxin A (OTA) pose serious health risks to humans and animals. Fruit pomace extracts, rich in natural nutrients and bioactive compounds, have the potential to enhance animal health and mitigate mycotoxin toxicity. This study evaluated a novel liquid anti-mycotoxin solution (LAS), a combination of grape and olive pomace extract administered to broiler chickens through drinking water (2 L:1000 L) for 1–42 days under a natural multi-mycotoxin challenge. The 42-day trial with 288 one-day-old male Ross 308AP95 chicks included four experimental groups: a negative control (NC); NC+LAS; a positive control (PC) group fed a diet containing 80 μg/kg AFs, 1600 μg/kg FBs, and 50 μg/kg OTA; and PC+LAS. The growth performance, oxidative defense genes (liver), and stress biomarkers (blood) were analyzed. Mycotoxin exposure negatively affected body weight (BW), the feed conversion ratio (FCR), and the oxidative defense mechanism. LAS supplementation improved BW and the FCR, reduced Nrf-2 expression, and enhanced mycotoxin detoxification via lower EPHX1 expression. Though the LAS did not fully restore performance to NC levels, it significantly mitigated mycotoxin-induced damage. This study concluded that the LAS is a promising solution to improve broiler resilience against moderate to high mycotoxin exposure.

## 1. Introduction

Mycotoxins are low molecular weight secondary metabolites produced under appropriate humidity and temperature conditions by filamentous fungi that pose serious risks to animal and human health [1]. With over 300 types of mycotoxins identified to date, mycotoxins can accumulate in various feeds and foods and make agricultural products unfit for consumption [2]. However, many mycotoxins remain poorly understood due to limited knowledge of their toxicokinetics and toxicity. Some well-studied examples of mycotoxins include aflatoxins (AFs), fumonisins (FBs), and ochratoxin A (OTA), all of which are of critical importance to both public health and agricultural economics [3]. Despite extensive research, the effects of these complex substances and their modified forms on poultry remain limited. The effects on poultry include reduced feed intake, growth performance, immunity, and antioxidant status, altered blood biochemical parameters, and increased mortality, carcinogenicity, teratogenicity, and organ damage [4,5,6]. These negative impacts not only affect animal health but also decrease productivity, resulting in economic losses for the poultry industry.

Aflatoxins are the most common type of mycotoxins produced by *Aspergillus flavus* and *Aspergillus parasiticus* [7]. Among its various types, Aflatoxin B1 (AFB1) is considered to be a widespread and highly potent toxin [8]. The consumption of AFB1-contaminated feed can result in aflatoxicosis in poultry and can thus reduce growth performance and immunity, alter blood biochemistry parameters and affect intestinal morphology in broilers [9,10,11]. AFB1 promotes the generation of free radicals and intensifies oxidative damage and lipid peroxidation, ultimately leading to cellular damage and death in both animals and humans [12,13]. Fumonisins, a group of over 28 mycotoxins, are produced by fungi of the *Fusarium* genus. Among them, fumonisin B1 (FB1) and fumonisin B2 (FB2) are the most common [14]. The adverse effects of FB1 in poultry can manifest in different ways, such as decreased weight gain, higher mortality rates, myocardial degeneration, myocardial hemorrhage, disruptions in the hemostatic mechanism, and hepatocyte necrosis [15]. Ochratoxin A (OTA), primarily produced by numerous species of *Aspergillus* and *Penicillium*, is a highly potent mycotoxin that poses significant risks to poultry health and productivity [16]. OTA impacts poultry by inhibiting mitochondrial function, increasing oxidative stress, and disrupting protein synthesis [17]. It also compromises intestinal barrier integrity by reducing tight junction protein expression, allowing harmful substances to penetrate the bloodstream [18]. OTA’s toxicity leads to growth impairments, immune suppression, and histopathological changes in organs, with chickens showing decreased body weight and elevated mortality rates at higher exposure levels. These adverse effects result in substantial economic losses in poultry farming [15].

In response to this challenge, feed additives such as mycotoxin adsorbents or binders, enzymes, probiotics, prebiotics, bacterial preparations, phytogenics, and antioxidants have emerged as a practical strategy to mitigate the harmful effects of mycotoxins in animal nutrition. These additives work by binding mycotoxins in the gastrointestinal tract to reduce their bioavailability, by enzymatically degrading them into less harmful compounds, or by boosting systemic health with natural extracts that are rich in phytogenics with antioxidant, anti-inflammatory, and hepatoprotective properties to mitigate the detrimental effects of mycotoxins [19,20]. Among various commercial products used for this purpose, the liquid anti-mycotoxin solution (LAS) is an additive for drinking water that is specifically formulated to enhance poultry performance under multiple mycotoxin challenges. By reducing oxidative stress and supporting the detoxification of mycotoxins, the LAS is designed to protect broilers from the adverse effects of naturally contaminated grains, thereby improving performance and overall flock health. It contains phytogenics from grape and olive pomace extracts, and technological additives such as preservatives and emulsifiers.

Several studies have explored the efficacy of grape (*Vitis vinifera*) extracts to ameliorate the toxicity of mycotoxins. Dietary supplementation with grape seed extract (GSE) effectively mitigates the toxic effects of AFB1 in broiler chickens. GSE reduced AFB1 residues in the liver and improved growth performance, immune response, liver function, and antioxidant enzyme activities. This suggests that GSE offers protective benefits against aflatoxicosis in poultry [21,22]. Another study concludes that grape seed extract effectively mitigates the harmful effects of fumonisin B1 (FB1) in broiler chickens. Supplementation with GSE at doses of 250 mg/kg and 500 mg/kg significantly improved growth performance, immune response, liver function, and antioxidant capacity, with the 500 mg/kg dose providing superior protection [23]. Similarly, the inclusion of grape pomace extract (GPE), which comes from the seed and skin parts of the fruit, has been shown to enhance thigh meat oxidative stability in broiler diets [24] and increase omega-6 polyunsaturated fatty acid (PUFA) levels [25]. A 2.5% grape pomace extract (GPE) diet improved gut morphology, enriched cecal microbiota composition, and optimized blood biochemical profiles in broilers, with no adverse effects on growth performance or meat quality [26]. Additionally, in ovo feeding using GPE at a 4 mg dosage significantly boosted hatchling growth, immune response, and antioxidant status, highlighting its potential as a natural antioxidant to enhance post-hatch performance and overall health [27]. Other promising phytogenic feed additives that show strong antimicrobial and antioxidant potential are olive (*Olea europea*) pomace extracts. The polyphenolic extract from olive pomace (OPE) showed antioxidant protection in cultured human HepG2 cells exposed to oxidative stress induced by tert-butylhydroperoxide [28]. Another study demonstrated that supplementing broiler chicken diets with 750 ppm of OPE enhanced growth performance, likely due to its anti-inflammatory properties. This was evidenced by reduced IL-8 expression and increased TGF-β4 and Bu-1 expression in the ileum [29]. Supplementation of 600 g dietary OPE/Ton showed a promising effect on the intestinal health of weanling piglets [30]. These findings suggest valuable industrial applications in food preservation, pharmaceuticals, and the natural antioxidants field.

This study aims to evaluate the efficacy of the LAS (a synergistic combination of grape and olive pomace extracts) administered through drinking water on performance parameters of broiler chickens exposed to nonspecific mycotoxin challenge as a result of feeding naturally mycotoxin-contaminated feeding grains. The biological effects of contaminated feeding grains were assessed by analyzing the activity of various molecular markers in the blood and vital tissues of male broiler chickens, both with and without LAS supplementation. Additionally, evaluating the impact of these additives on immune responses and nutrient absorption is crucial for determining their overall efficacy. By addressing these aspects, this study seeks to provide valuable insights into the practical application of the novel liquid anti-mycotoxin solution (LAS) in commercial poultry production and its potential to improve broiler resilience in environments prone to mycotoxin contamination.

## 2. Results

### 2.1. Diet Composition

The chemical composition of the experimental diets is presented in Table 1. The results showed that the chemical composition of the basal diets from all three phases was within the normal ranges of variation and closely met the specifications of the diet for Ross 308AP95 chicks in terms of DM, OM, CP, EE, and CF. The total phosphorus and calcium contents were within the normal ranges of variation.

Concentrations of AFs, FBs, and OTA in the starter, grower, and finisher diets of the NC and the PC groups are presented in Table 1. The NC group exhibited negligible concentrations of AFs and OTA across all stages. In the PC group, AF concentrations were nearly four times higher than the EU regulatory limit of 20 μg/kg at all stages [31]. While OTA levels were elevated, they remained below the EU threshold of 100 μg/kg [32]. FB concentrations were negligible compared to the EU limit of 20,000 μg/kg. Overall, the diet used in this study posed a significant challenge, particularly due to high AF concentrations, while the OTA levels presented a moderate challenge.

### 2.2. The Temperature and Relative Humidity of the Experimental House

The temperature and relative humidity of the experimental house are shown in Figure 1a,b, respectively. During the first seven days of the experiment, the mean temperature varied around 30 °C, after which it was maintained at approximately 27–28 °C for the rest of the study. The gap between the maximum and the minimum temperature was wider at the beginning, which was narrowed down as the experiment progressed. The relative humidity (RH) was fairly constant and varied between 85 and 90% throughout the experiment, with some dips being observed in the minimum RH at the middle of the experiment.

### 2.3. Growth Performance

The data related to the performance traits of the experimental birds fed with the mycotoxin-contaminated diets and supplemented with the LAS are presented in Table 2 and Figure 2, Figure 3 and Figure 4.

**Effect of feeding contaminated diets:** Feeding using the contaminated diet had a significant effect on the BW of the birds after 24 d (main effect contamination *p* = 0.022) and 42 d (main effect contamination *p* < 0.0001). Growth depression due to feeding using the contaminated diet could be seen in the birds during 11–24 d, as could be evidenced from the lower ADG in the birds fed with the contaminated diet (main effect contamination *p* = 0.014) during this period of life, and the depression became more severe during 25–42 d and 1–42 d (main effect contamination *p* < 0.001). The feed intake varied between the treatment groups, albeit without any specific trend with regard to the presence of the mycotoxin-contaminated diet. Feeding using the contaminated diet had a negative effect on the ADFI (Figure 2a,b) of the birds during 25–42 d and 1–42 d (main effect contamination *p* = 0.001). Mycotoxin contamination of the diet had little effect on the FCR of the birds during 1–10 d (main effect contamination *p* = 0.103). The FCR worsened (Figure 3a,b) due to mycotoxin contamination during 11–24 d (main effect contamination *p* = 0.008), 25–42 d (main effect contamination *p* = 0.001), and 1–42 d (main effect contamination *p* < 0.0001). Feeding using the contaminated diet had no effect on the liveability of the birds (*p* > 0.05). However, the EPI deteriorated when the birds were exposed to the contaminated diets (main effect contamination *p* < 0.0001).

**Effect of LAS supplementation:** Supplementation using the LAS improved the BW (Figure 4a,b) of the birds after 24 d (main effect LAS *p* = 0.003) and 42 d (main effect LAS *p* = 0.011). Supplementation using the LAS had a positive effect on the growth rate of the birds, irrespective of the mycotoxin contamination of the diets during 1–10 d (main effect LAS *p* = 0.003), 11–24 d (main effect LAS *p* = 0.012), and 1–42 d (main effect LAS *p* = 0.011). Supplementation using the LAS increased the ADFI during 1–10 d in the birds fed with the contaminated diet (main effect LAS *p* = 0.002) and tended to increase the same manner during 11–24 d in the birds fed with the standard as well as the contaminated diets (main effect LAS *p* = 0.062). The effect of the LAS on the ADFI was not conspicuous during 25–42 d and 1–42 d (main effect *p* > 0.05). Supplementation using the LAS had no effect on the FCR of the birds during 1–10 d and 11–24 d (main effect LAS *p* > 0.05). During 25–42 d, LAS supplementation improved the FCR of the birds (main effect LAS *p* = 0.045), and this trend continued when the data were pooled together for the period of 1–42 d (main effect LAS *p* = 0.001). Supplementation using the LAS had a positive effect on the EPI of the birds (main effect LAS *p* = 0.006).

**Interaction effects:** It was observed that although a contamination*LAS interaction was lacking with regard to the BW on 24 d (*p* = 0.437), a trend could be observed (*p* = 0.067), which suggested that supplementation using the LAS might have had worked better in matured birds than the younger birds in the presence of mycotoxin contamination and resulted in a better BW as compared with the untreated birds. The growth rate of the birds did not show any significant contamination*LAS interaction, except during 1–42 d (*p* = 0.067) which suggested that the LAS had an overall positive effect on the growth rate of the experimental birds, but the data were not robust enough to indicate a definitive positive effect of the LAS when the contaminated diets were offered to the birds in general. No contamination*LAS interaction could be observed with regard to the FCR of the birds during 1–10 d and 11–24 d, although the interaction was significant during 25–42 d (*p* = 0.045). The trend was continued when the data were pooled together over the period of 1–42 d (*p* = 0.02). However, no contamination*LAS interaction could be observed with regard to liveability (*p* = 0.757) and the EPI (*p* = 0.206) of the experimental birds.

#### Carcass Traits

The gross carcass traits and the weight of the internal organs (the absolute weight and weight relative to the live BW) are presented in Table 3.

**Effect of feeding contaminated diets:** The dressed carcass weight was not affected by feeding the contaminated diets to the birds (main effect *p* = 0.729). The weights of the liver, heart, gizzard, giblets, abdominal fat, and that of the commercial cuts (breast, drumstick, and thighs) were also not affected by feeding the contaminated diets to the birds (*p* > 0.05). The weight of the aforementioned internal organs relative to the live weight was similar across the level of mycotoxin contamination, and hence it was inferred that the mycotoxin contamination employed in this study did not influence the absolute and relative weights of the internal organs and commercial cuts of the chickens (*p* > 0.05).

**Effect of LAS supplementation:** Supplementation using the LAS had no effect on the absolute and relative weights of the internal organs and the commercial cuts (*p* > 0.05).

**Interaction effects:** No interaction between the level of contamination and supplementation using the LAS could be observed in this experiment with regard to the absolute weight of the internal organs (*p* > 0.05) and the weight of the same relative to BW (*p* > 0.05).

### 2.4. Hepatic and Antioxidant Biomarkers in Serum

The concentration and activity of hepatic and antioxidant biomarkers in serum are presented in Table 4.

**Effect of contaminated diet:** There was no apparent effect of feeding using the mycotoxin contaminated diet on the concentration and activities of GSH-Px, SOD, ALP, ALT, and γ-GT after 10 d (main effect *p* > 0.05). The activities of GSH-Px, ALP, AST, and ALT did not vary due to feeding using the contaminated diets after 24 d (main effect *p* > 0.05). Feeding using a contaminated diet led to variations in SOD activity at 24 days (main effect: *p* = 0.001), though no clear trend was observed. Serum AST activity varied between the groups at 10 days of age due to feeding using contaminated diets (main effect: *p* = 0.034); however, no such effect was observed at 24 days (main effect: *p* = 0.594). Additionally, γ-GT activity was significantly affected by the contaminated diet at 24 days (main effect: *p* = 0.01) and appeared higher in birds that were fed the contaminated diets.

**Effect of LAS supplementation:** Supplementation using the LAS did not have any significant effect on any of the marker concentrations/activities either at 10 d or at 24 d of age (main effect *p* > 0.005).

**Interaction effects:** The interaction effects were mostly insignificant, except for SOD at 24 d (*p* = 0.001). A trend was observed for γ-GT at 24 d (*p* = 0.077). For SOD, the interaction effect showed that LAS supplementation decreased SOD activity in birds fed the standard diet, whereas it increased SOD activity in those on the contaminated diet.

Overall, ANOVA revealed that LAS supplementation reduced SOD activity in birds fed the standard diet (NC+LAS) but increased it in those on the contaminated diet (PC+LAS) compared to the NC and PC groups (*p* < 0.001). At 10 days, AST activity was lowest in the NC group across all treatments, while LAS supplementation led to significantly higher AST activity in the PC+LAS group (*p* < 0.05). The activity of γ-GT was also lower in the NC group on 24 d as compared to the PC and the PC+LAS groups (*p* < 0.05). Overall, the present data indicated mild to moderate effects of feeding the contaminated grains on the activities of different biomarkers. It is likely that the mycotoxin challenge imposed on the birds was not severe enough to cause significant changes in hepatic function and antioxidant activity. As a result, any potential effects of the LAS remained largely undetectable.

### 2.5. Expression of Nrf-2 and EPHX1 Genes in Hepatic Tissues

The expression of Nrf-2 and EPHX1 measured at 10 and 24 d of age relative to the NC group (fold change) is presented in Figure 5a and Figure 5b, respectively. The relative fold change of Nrf-2 gene (Figure 5a) at 10 d was higher in both the PC and PC+LAS groups as compared to the NC and NC+LAS groups (main effect contamination *p* = 0.011; main effect LAS *p* = 0.001; ANOVA *p* < 0.05), which suggested that there was an overall upregulation in the expression of Nrf-2 activity due to feeding using the contaminated diet and supplementation using the LAS, especially in younger birds. At 24 d, the relative fold change was significantly higher in the PC group as compared to that in the NC, NC+LAS, and PC+LAS groups (main effect contamination *p* = 0.038, main effect LAS *p* = 0.001, contamination*LAS interaction *p* = 0.016).

The relative fold change for the EPHX1 gene at 10 d of age was lower in the PC+LAS group as compared to the PC group. It should be noted that the expression of the same gene on 10 d in the former group was significantly higher than that in the NC and NC+LAS groups (main effect contamination *p* = 0.144, main effect LAS *p* = 0.001, contamination*LAS *p* = 0.078). At 24 d, the relative fold change was higher in both the PC and the PC+LAS groups as compared to the NC and NC+LAS groups (main effect contamination *p* = 0.219, main effect LAS *p* = 0.001, contamination*LAS *p* = 0.915).

## 3. Discussion

Extensive research over the past decades has confirmed the widespread prevalence of mycotoxins in most feed ingredients. A global survey in 2013 reported that 81% of approximately 3000 grain and feed samples contained at least one mycotoxin, an increase from the 10-year average of 76% (2004–2013) [33]. This rise is likely due to advancements in detection technologies, particularly in high-performance liquid chromatography coupled with tandem mass spectrometry, which is highly selective, sensitive, and accurate [34]. Mycotoxins can harm animals individually or synergistically, affecting organs like the gastrointestinal tract, liver, and immune system and leading to reduced productivity or even mortality in severe cases. While mycotoxin-binding agents are widely used in animal production, the chemical diversity of mycotoxins makes it clear that no single method can effectively deactivate them all. A combination of nutritional strategies, including enzymatic or microbial detoxification (biotransformation/biodetoxification), is required to neutralize specific mycotoxins without compromising feed quality. Despite the growing availability of modern detection and detoxification techniques, awareness among feed producers about mycotoxin prevalence, their harmful effects, and safe mitigation strategies remains limited.

The findings of this study indicate that feeding broiler chickens a diet contaminated with mycotoxins adversely affects their body weight (BW) and feed conversion ratio (FCR). This observation aligns with previous research that has demonstrated the detrimental effects of mycotoxins on poultry performance [35,36,37]. However, supplementation using the liquid anti-mycotoxin solution (LAS) notably improved the BW and FCR of birds that were fed the contaminated diet, suggesting a mitigating effect on the adverse impacts of mycotoxins in poultry. This is consistent with previous findings that showed the positive impact of the two main components of the LAS, grape and olive pomace extracts, as anti-mycotoxin agents. Grape pomace is a rich source of valuable compounds, including unsaturated lipids, sterols, vitamins, antioxidants, fiber, volatile organic compounds, and phenolic compounds. Phenolic compounds are notable for their health benefits, such as reducing oxidative stress, inflammation, and exhibiting anticancer properties [38]. Key phenolic compounds in grape pomace extract (GPE) include simple phenols like hydroxycinnamic and hydroxybenzoic acids, as well as polyphenols such as tannins, stilbenes, and flavonoids [39]. GPE has demonstrated bio-fungicidal properties against major mycotoxigenic species, effectively inhibiting OTA production by 2–57% and AFB1 production by 5–75% [40]. Studies on livestock have highlighted the effectiveness of grape seed extract (GSE) in mitigating mycotoxin-related damage. In one-day-old broiler chickens, supplementation with 250–500 mg/kg of GSE reduced oxidative stress in the liver and plasmatic cytokine levels caused by 400 mg/kg of FB1. It also improved jejunal morphology, ileal microbiota balance, BW, the FCR, and mortality rates, enhancing overall growth performance [23]. Additionally, GSE alleviated the adverse effects of AFB1 on immune function, antioxidant capacity, and liver health by reducing AFB1 residues and liver damage. This resulted in increased serum immunoglobulin levels, enhanced antioxidant parameters, and better growth outcomes, including improved relative organ weights, ADG, and the FCR [21,41]. In quails, supplementation with 500 mg/kg of GSE alleviated the harmful effects of 1 mg/kg of AFB1, such as oxidative stress, lipid peroxidation, liver fibrosis, and sinusoidal dilation, while also improving the FCR, BW, and BWG [42]. In weaned piglets exposed to 320 µg/kg of AFB1, an 8% grape seed meal reduced oxidative stress and inflammatory markers in the mesenteric lymph nodes, spleen, and liver [43,44]. These findings suggest that grape pomace extract, which is manufactured from the seed and skin of the fruit, is a cost-effective solution for mitigating mycotoxin contamination in feed, offering antioxidant and antimicrobial properties and serving as a promising anti-mycotoxin agent.

Another component of the LAS, olive pomace, is a rich source of bioactive phytochemicals, including vitamin E vitamers, phenolic compounds, peptides, quercetin, and fats. Among these, some phenolic compounds are particularly significant, with hydroxytyrosol being the most abundant, followed by tyrosol and oleuropein. Hydroxytyrosol, a potent natural antioxidant derived from oleuropein hydrolysis, exhibits a wide range of biological activities, including antioxidant, antitumor, anti-inflammatory, and antimicrobial effects. It also plays a role in regulating lipid profiles and protecting against genotoxicity, cytotoxicity, and proapoptotic effects [45,46]. In the context of mycotoxin mitigation, olive pomace demonstrates antifungal properties and can reduce mycotoxin levels. It has been shown to inhibit the growth of toxigenic fungi and decrease aflatoxin levels [47]. An in vitro study highlighted its ability to adsorb AFB1, zearalenone (ZEN), and OTA mycotoxins at 30 mg/mL across various pH levels [48]. Specifically, hydroxytyrosol protects against ochratoxin A (OTA)-induced oxidative stress in rat kidneys by reducing malondialdehyde (MDA), lactate dehydrogenase (LDH), and reactive oxygen species (ROS) release [49]. Similarly, olive mill wastewater has been reported to inhibit AFB1 production [50]. Beyond its mycotoxin-mitigating potential, olive pomace also benefits animal health. In one-day-old broiler chickens, it reduced gut inflammation by downregulating pro-inflammatory cytokines while upregulating anti-inflammatory cytokines [29]. Quercetin, another component of olive pomace, has demonstrated protective effects against T-2 toxin-induced liver damage in tilapia, enhancing antioxidant capacity, improving liver function, and increasing survival rates, weight gain, and the hepatosomatic index [51]. This multifaceted functionality underscores the potential of olive pomace as a valuable bioresource for mitigating mycotoxins and supporting health in various applications.

Interestingly, this study found that gross carcass traits and the weights of internal organs relative to live weight were not significantly affected by either the contaminated diet or LAS supplementation. This contrasts with some of the literature, which suggests that mycotoxin exposure can lead to organ damage and altered carcass characteristics [52]. The lack of significant effects on these parameters in our study may indicate a threshold level of mycotoxin exposure below which organ integrity is preserved or suggest an effective protective role of LAS supplementation.

Regarding oxidative stress indicators, the observed decrease in superoxide dismutase (SOD) activity with LAS supplementation in the NC group and its increase in the PC group provides insight into the modulatory effects of LAS on the antioxidant defense mechanisms in chickens. Enhanced SOD activity in response to oxidative stress is a common physiological response; therefore, the modulation of this enzyme’s activity indicates the potential for LAS to influence the oxidative status of birds [53]. Moreover, the upregulation of the Nrf-2 gene expression in birds that were fed the contaminated diet suggests a compensatory response to oxidative stress, which aligns with previous research highlighting the role of Nrf-2 in cellular defense against oxidative damage [54]. The finding that LAS supplementation mitigated the upregulation of the Nrf-2 gene supports the notion that the LAS may enhance the oxidative defense mechanism in broilers. This is significant as it highlights the potential of LAS to not only alleviate the negative performance impacts associated with mycotoxin exposure but also to support the birds’ inherent antioxidant defenses [55]. Furthermore, the decreased expression of the EPHX1 gene in birds supplemented with LAS suggests a possible reduction in the biotransformation of mycotoxins to less harmful metabolites, indicating that the LAS may modulate metabolic pathways related to mycotoxin detoxification. This observation warrants further investigation into the specific mechanisms by which LAS interacts with these pathways and its overall impact on mycotoxin metabolism in poultry.

Overall, this study underscores the importance of dietary management in poultry production, particularly concerning the risks posed by mycotoxin contamination. This study demonstrates that the supplementation of the novel liquid anti-mycotoxin solution (LAS) in drinking water effectively mitigates the adverse effects of moderate to high mycotoxin challenges in broiler chickens, improving BW and the FCR while augmenting the oxidative defense mechanisms of the birds. Indeed, the decrease in SOD and increase in γ-GT induced by mycotoxin contamination was prevented by the LAS. Based on the observation, the LAS most likely protects against mycotoxin-induced damage by neutralizing oxidative stress and supporting liver detoxification. Mycotoxins generate excessive reactive oxygen species (ROS), depleting antioxidant enzymes like SOD and overloading hepatic detoxification, as shown by increased gamma-glutamyl transferase (γ-GT). This decreases ROS production, preserving SOD activity and reducing the burden on liver detoxification pathways, which stabilizes γ-GT levels. As a result, the LAS mitigates oxidative damage and restores antioxidant and hepatic function, supporting poultry health.

The findings support the use of the LAS as a valuable intervention to enhance poultry performance under mycotoxin challenges, particularly at moderate to high exposure levels. Future research should focus on elucidating the molecular mechanisms through which the LAS exerts its protective effects and exploring its efficacy across different strains of poultry and under varying environmental conditions.

## 4. Conclusions

The present study established that feeding using a mycotoxin-contaminated diet with significantly high concentrations of AFs, FBs, and OTA negatively impacted BW and the FCR in broiler chickens. However, supplementation with the LAS resulted in an economically considerable improvement in BW and the FCR, particularly in birds exposed to a mycotoxin-contaminated diet. While growth performance was affected by mycotoxin exposure, gross carcass traits and internal organ weights relative to the live weight remained unaffected, regardless of diet composition or LAS supplementation.

In regards to the oxidative stress response, LAS supplementation decreased superoxide dismutase activity in birds fed with the non-contaminated diet but increased in the mycotoxin-contaminated diet. This study also indicated that feeding the contaminated diet negatively affected the oxidative defense mechanism, as evidenced by the upregulated expression of Nrf-2. LAS supplementation appeared to mitigate this effect by reducing Nrf-2 expression. Similarly, the expression of EPHX1, which plays a role in the detoxification of mycotoxins, was comparatively lower in LAS-supplemented birds, suggesting a protective effect of the LAS against mycotoxin-induced stress.

Overall, it was concluded from the present study that supplementation of the liquid anti-mycotoxin solution (LAS) at a concentration of 2 L:1000 L for 1–42 d in broiler chickens was capable of alleviating the negative effects of a natural multi-contaminated diet by high levels of AFs, OTA, and moderate levels of FBs. Although the BW and FCR of the supplemented birds fed with the contaminated diet were not comparable with those of the birds fed with the standard diet, a substantial improvement may be possible with LAS supplementation in cases of impending moderate to high levels of mycotoxin challenge.

## 5. Materials and Methods

### 5.1. The Outline of the Experiment and a Description of Dietary Treatments

The current study was specifically designed for commercial broiler production, where birds are typically marketed or processed by 42 days of age. The experiment was conducted on a flock of 288 male Ross 308AP95 chicks, tested over a period of 1–42 days. The chicks, sourced from a commercial hatchery, were reared on litter in pens throughout the study. They were assigned to treatment groups and distributed across pens using a completely randomized design, with four treatment groups, each comprising six replicate pens and 12 chicks per pen at the experiment’s start.

The test component used in this study i.e., the LAS (liquid anti-mycotoxin solution) is composed of grape pomace extract (rich in polyphenolic compounds such as proanthocyanidins, anthocyanins, procyanidins, catechin, ferulic acid, quercetin, resveratrol, and syringic acid, among other bioactive compounds and derivates) and olive pomace extract (rich in hydroxytyrosol, tyrosol, and other polyphenolic derivates). In addition, the LAS contains citric acid, sorbic acid, water, and sorbitol of high purity as chemical components. In vitro tests were conducted to evaluate the efficacy of LAS at various dosages, identifying a suitable range of 0.5 to 2 L per 1000 L of drinking water for poultry. As this study represents the first scientific in vivo trial of the LAS in broilers, only the maximum recommended dosage was tested to enable the clear observation of the product’s effects when administered through drinking water.

The chicks were fed with four types of diets: First, a negative control (NC) diet, formulated with maize in which the concentrations of some of the major mycotoxins, viz., aflatoxins (AFs), fumonisins (FBs), and ochratoxin A (OTA) were below the incriminating levels (based on European regulations). Second, the NC group supplemented with the liquid anti-mycotoxin solution (NC+LAS) had access to drinking water enriched with the LAS at a concentration of 2 L:1000 L throughout the study, i.e., from day 1 to day 42. Third, a positive control (PC) diet, which included maize contaminated with these mycotoxins (AFs, FBs, and OTA) at sufficient levels to elevate the overall mycotoxin loads in the final diet to incriminating levels. Similar to NC+LAS, a fourth diet named PC+LAS where the experimental birds were fed with the PC diet but had access to drinking water supplemented with LAS at a concentration of 2 L:1000 L ad libitum throughout the study.

Naturally contaminated maize, maize gluten meal, and groundnut meal were used as the source of mycotoxins in this study. The ingredients were checked for concentrations of AFs, FBs, and OTA toxins. However, it was only AFs and OTA that were found in incriminating concentrations, while FB levels were moderately high. The concentration of AFs was 117 μg/kg and 122 μg/kg in maize and maize gluten meal, respectively; the concentration of OTA in maize gluten meal was 149 μg/kg, and the FB concentration in the same material was 800 μg/kg. The concentrations of OTA in maize and maize gluten meal were 8 μg/kg and 11 μg/kg, respectively. The concentration of AFs and FBs in the groundnut meal was 100 μg/kg and 250 μg/kg; and the contaminated maize also contained 6000 μg/kg of FBs. All the aforementioned ingredients were utilized to formulate the NC and PC diets, ensuring that both diets were iso-caloric and iso-nitrogenous with minimal variations in nutrient composition. This approach was taken to confirm that any observed differences in performance were attributable to the effects of feeding the contaminated diets and the supplementation of the LAS.

### 5.2. Chemical Analysis of Diet

Multiple samples for each diet were collected, pooled, and analyzed for dry matter (DM, method number 934.01), nitrogen and crude protein (CP, method number 954.01), ether extract (EE, method number 920.39), crude fiber (CF, method number 962.09/AOAC 978.10), total ash (AOAC 942.05-2012) and organic matter (OM, method number 942.05), calcium, and total phosphorus (AOAC 927.02-1990 and AOAC 965.17-1966).

General bird husbandry and measurement of performance traits

Upon arrival at the experimental farm, the chicks were weighed (mean BW 44.6 g) and evenly distributed according to a completely randomized design. The chicks were raised in pens (1.2 m × 1.2 m) on litter composed of wood shavings. The lighting schedule was 23 h during the first 7 days, followed by 20 h a day until harvest. A three-phase feeding was practiced in which the starter (1–10 d) diet was given as crumbles, and the grower (11–24 d) and the finisher (25–42 d) diets were given as pellets. The composition of the basal diet is given in Table 5. The birds received feed within 12 h of hatching. Manually operated feeders and drinkers were fitted to each pen, and the birds had ad libitum access to feed and water throughout the duration of the experiment. Vaccination was provided against infectious bronchitis (0 d), Newcastle disease (5 d and 20 d), and infectious bursal disease (12 d). The body weight (BW) was recorded pen-wise at 10 d, 24 d, and 42 d at the same time of day (08:00 h) without any fasting. A measured quantity of feed was offered daily to each of the pens in two equal divisions. The cumulative feed intake (FI) was calculated on a weekly basis by subtracting the quantity of feed left in each pen from the total quantity of the feed offered during the preceding week. The average daily BW gain (ADG) and average daily feed intake (ADFI) were calculated for 1–10 d, 11–24 d, 25–42 d, and 1–42 d, and the feed conversion ratio (FCR) was calculated as the ADFI/ADG ratio for the corresponding feeding periods. The test facility, pens, and birds were observed twice daily for general flock condition, lighting, water, feed, ventilation, and unanticipated events. Starting from the beginning, any bird that was culled, found dead, or sacrificed was recorded. All mortalities were subjected to necropsy to determine the probable cause of death. The weight of the dead birds was recorded to calculate the mortality-corrected FCR. The European Productivity Index (EPI) was calculated as: EPI = [(100 − mortality) × (mean BW (kg)/age) × 100]/FCR.

### 5.3. Mycotoxin Analysis of Raw Materials and Diets

The mycotoxin levels in raw materials and complete feed were measured using water extracts. The analysis was based on an immunoreceptor assay with lateral flow technology, utilizing the MycoMaster system (Trouw Nutrition, Amersfoort, The Netherlands). However, for determining the concentration of OTA, water extraction was unsuitable. Therefore, 70% methanol was used to extract the required amount of toxin from the raw materials and complete feed. Specific test strips were used for the quantitative detection of each mycotoxin: the AFQ-WETS5 strip (Charm Sciences Inc., Lawrence, MA, USA) for AFs (using water extraction), the FUMQ-WET5S strip (Charm Sciences Inc., Lawrence, MA, USA) for FBs (using water extraction), and the OCHRAQ strip (Charm Sciences Inc., Lawrence, MA, USA) for OTA (using methanol extraction). For maize and complete feed, an extraction procedure of 1:3 dilution was followed as per the manufacturer’s protocol for the estimation of AFs and FBs. Briefly, the ground sample (50.0 ± 0.1 g) was weighed into a beaker with a closable lid, followed by the addition of 1–2 sachets of extraction powder and 150 mL of distilled water. The mixture was shaken firmly for 1.5 min and allowed to settle. The clear liquid was then transferred to a microcentrifuge tube and centrifuged for 20 s within 30 min of extraction. The centrifuged liquid was filtered through an RC15 filter (0.45 μm) into a clean microcentrifuge tube to obtain the extract. For analysis, the extract was diluted with the appropriate dilution buffer provided with the test strips, depending on the toxin being analyzed, within 2 h of centrifugation. The diluted solution was mixed thoroughly by shaking or vortexing for 5 s before proceeding with further analysis. For the groundnut meal and the maize gluten meal, the same process was followed; however, the dilution ratio was 2:5. Finally, 300 μL of the diluted solution was added onto the respective strips and placed in the MycoMaster system, and the data were recorded.

For the analysis of Ochratoxin A, an extraction procedure of 1:2 dilution was followed as per the manufacturer’s protocol. Briefly, the ground sample (50.0 ± 0.1 g) was weighed into a closable beaker, and 100 mL of 70% methanol was added. The mixture was shaken firmly for 1 min and allowed to settle. A 2 mL aliquot of the clear liquid was transferred to a microcentrifuge tube and centrifuged for 20 s. The centrifuged liquid was filtered through an RC15 filter (0.45 μm) into a clean microcentrifuge tube and is referred to as “the extract”. To 100 μL of the extract, 1000 μL of the dilution buffer was added within 2 h of centrifugation. The diluted solution was thoroughly mixed by shaking or vortexing for 5 s before analysis. Finally, 300 μL of the diluted solution was added onto the respective strips and placed in the MycoMaster system (Trouw Nutrition, Amersfoort, Netherlands), and the data were recorded.

### 5.4. Collection and Analysis of Blood

At 10 and 24 days of age, one bird was randomly selected from each replicate pen (n = 6 per treatment). The whole blood was collected from the right brachial vein using vacutainer tubes without any anticoagulant. The serum was then separated by centrifugation at 2500 × g for 10 min and stored at −20 °C for later analysis of oxidative stress biomarkers like superoxide dismutase (SOD), reduced glutathione (GSH-Px), and hepatic tissue damage markers such as gamma-glutamyl transferase (γ-GT). Biomarkers were quantified using chicken-specific ELISA kits (Bioassay Technology Laboratory, Shanghai, China/DRG Instruments GmbH, Marburg, Germany) with a microplate reader (Biotek ELX 800 TS ELISA reader, BioTek Instruments, Winooski, VT, USA). An additional serum aliquot was analyzed for liver function biomarkers, including aspartate aminotransferase (AST), alanine aminotransferase (ALT), and alkaline phosphatase (ALP). Biochemical kits from Delta Lab, India, were used for the analysis. AST and ALT were analyzed following the principle of the Optimized IFCC method, and ALP was analyzed following the principle of PNPP/AMP. For AST and ALT, 100 μL of the serum sample was added to 1000 μL of the specific working reagent, and after 60 s of incubation, the decrease in absorbance every minute during 3 min at 37 °C was measured at 340 nm. For ALP, 20 μL of the serum sample was added to 1000 μL of the working reagent, and after 60 s incubation, the change in absorbance every 30 s for 90 s at 37 °C was measured at 405 nm. These analyses were performed photometrically in a semi-automated blood biochemistry analyzer (Rayto RT-9200, Diatek Healthcare Pvt Ltd, Kolkata, WB, India) using commercial kits following the manufacturer’s protocol, as mentioned earlier.

### 5.5. Expression Analysis of Hepatic Function Biomarker Genes

This study evaluated whether LAS supplementation could enhance birds’ antioxidative defenses by assessing nuclear factor erythroid-2 related factor-2 (Nrf-2) and Epoxide Hydrolase 1 (EPHX1) gene expression in liver cells, hypothesizing that the LAS might mitigate mycotoxin-induced liver impairment. The mRNA expression was studied using 8 random samples from each treatment at 10 and 24 d of age. The liver was eviscerated, washed with phosphate-buffered saline (PBS), and approximately 100 mg of tissue sample was excised for mRNA expression analysis of Nrf-2 and EPHX1 genes in the liver. The tissue samples were stored at –80 °C in RNAlater^®^ (Sigma-Aldrich, Bangalore, India) until use for further processing. The tissue segments were homogenized together in liquid nitrogen to yield a composite tissue, from which the RNA was extracted. The total RNA from the tissues was extracted using Trizol (Ambion, Thermo Fisher Scientific, Waltham, MA, USA). Up to 2 µg of the total RNA was taken for genomic DNA (gDNA) elimination using gDNA Wipeout Buffer. The tubes were incubated at 42 °C for 2 min and then were immediately placed on ice. The complementary DNA (cDNA) was synthesized using a cDNA reverse transcription kit (QuantiTect Reverse Transcription Kit, Qiagen Inc., Hilden, Germany).

The primer sequence used for the target and housekeeping genes is provided in Table 6. Real-time PCR analysis was performed on Bio-Rad CFX-96 real-time PCR instrument (Bio-Rad Laboratories Inc., Hercules, CA, USA). Each reaction contained 12.5 ng RNA equivalents, 200 to 250 nmol/L of forward and reverse primers for each gene, and 10 µL of GoTaq qPCR Master Mix (GoTaq^®^ qPCR Master Mix, Promega, Madison, WI, USA) for a final volume of 20 µL. The PCR reaction conditions were as follows: 95 °C for 3 min for initial denaturation, followed by 40 cycles of extension at 95 °C for 30 s and the primer-specific annealing temperature for 1 min. Gene quantification used the 2^−ΔΔCt^ method, with GAPDH as the housekeeping gene. The expression levels of the genes were presented as fold change increments (2^−ΔΔCt^) in the treatment group supplemented with the LAS compared to the untreated control group.

### 5.6. Measurement of Gross Carcass Traits

At the end of the feeding trial on 42 d, 1 bird from each of the replicates (6 birds from each of the treatments) were selected randomly to study the gross carcass traits. The birds were killed by mechanical stunning followed by exsanguination. The hot carcass was scalded (at 55 °C with intermittent dipping for 20 s at a time) and picked up using de-feathering equipment. The carcass was eviscerated and the weight of the eviscerated carcass along with that of the internal organs (heart, liver, and gizzard) and abdominal fat were expressed relative to the live weight of the killed birds. The weight of the breast, legs and drumsticks were expressed relative to live weight. The Pectoralis major muscle was extracted from the leg and was subjected to analysis of nitrogen (protein calculated as N × 6.25) according to AOAC Official Method (2011.04) and crude fat (according to Gerhardt Analytical Methods).

### 5.7. Statistical Analysis

All data were analyzed statistically by 2 × 2 factorial ANOVA, with PROC GLM of SAS 2012 (On Demand for Academics) using the diet type (standard or contaminated) as the main factor. For all performance data, the replicates were considered as the experimental units, while for the rest of the parameters the individual observations served as the experimental units. Probability values of *p* < 0.05 were described as statistically significant and those of *p* < 0.1 were described as a trend. The following statistical model was used for analyzing the data:

Yijk = μ + αi + βj + (α*β)ij + εijk where, Yijk is the response variable, μ is the overall mean, αi is the effect of ith level of contamination, βj is the effect of jth level of LAS in diet, (α*β)ij is the interaction effect between the ith level of factor α (contaminated diet) and jth level of factor β (LAS), and εijk is the error term present in the model.

The BW and other performance data were plotted in a violin plot, to visualize the summary of data distribution and the probability density of the data. The scatter and the violin plots, made using python programming language (Google Co-laboratory Notebook), included the minimum, first quartile, median, third quartile, and maximum values.

## Figures and Tables

**Figure 1 toxins-17-00212-f001:**
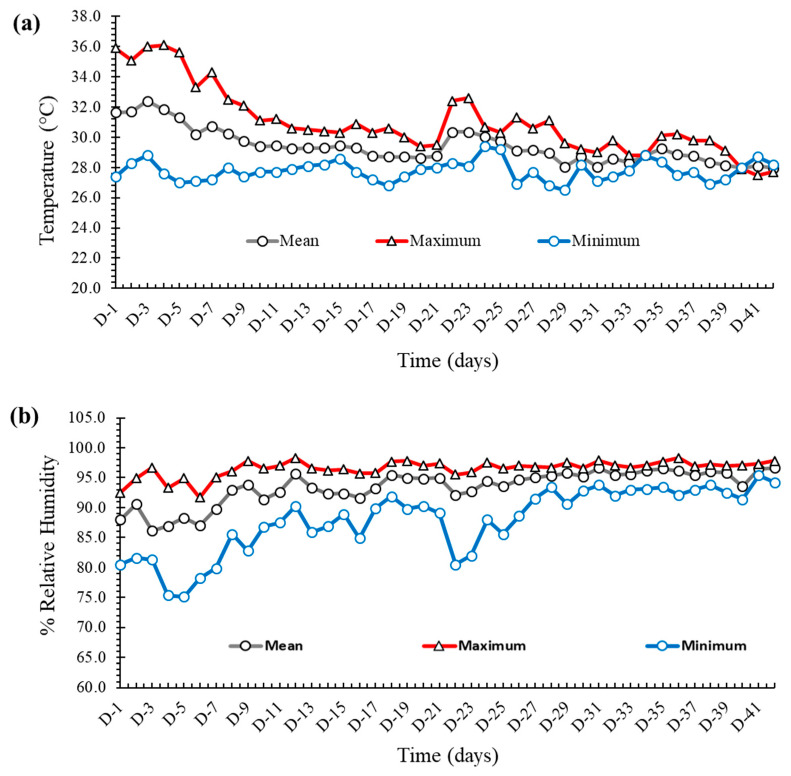
(**a**) The maximum, minimum, and mean temperature (values in °C on the Y axis) of the experimental house. (**b**) The maximum, minimum, and mean relative humidity (values in % on the Y axis) of the experimental house.

**Figure 2 toxins-17-00212-f002:**
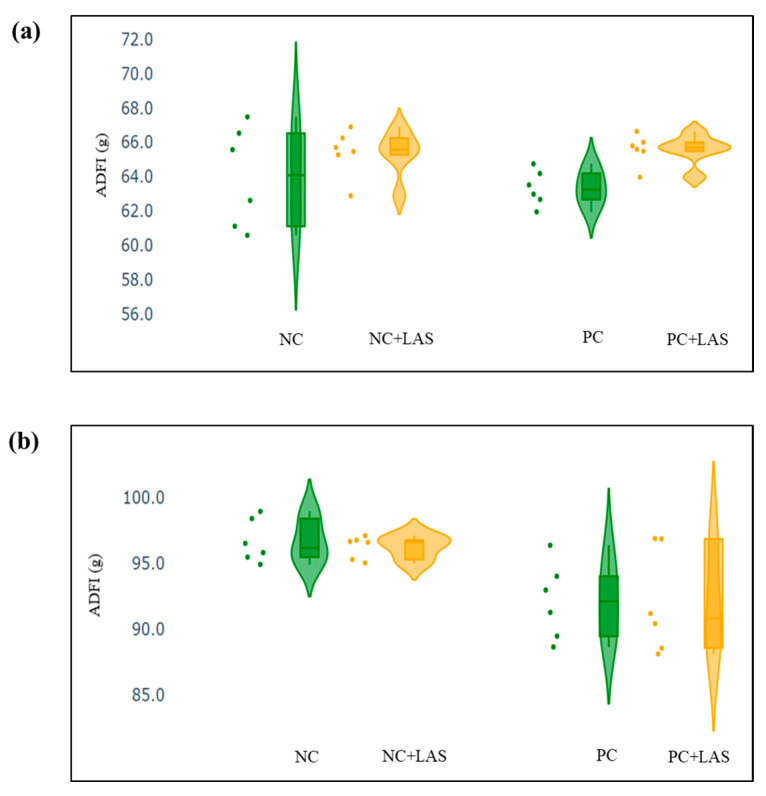
(**a**) Scatter and violin plot for average daily feed intake (ADFI) during 1–24 d of age. (**b**) Scatter and violin plot for ADFI during 1–42 d of age.

**Figure 3 toxins-17-00212-f003:**
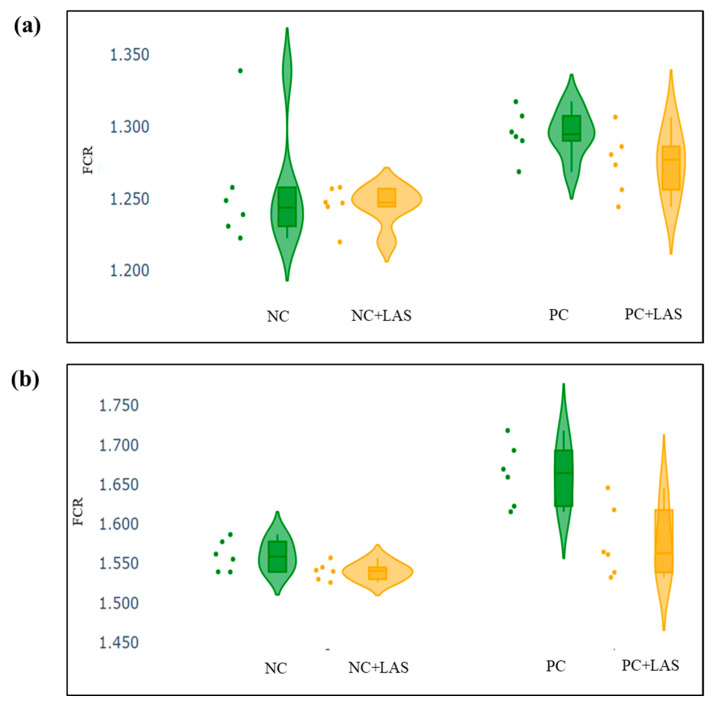
(**a**) Scatter and violin plot for feed conversion ratio (FCR) during 1–24 d of age. (**b**) Scatter and violin plot for FCR during 1–42 d of age.

**Figure 4 toxins-17-00212-f004:**
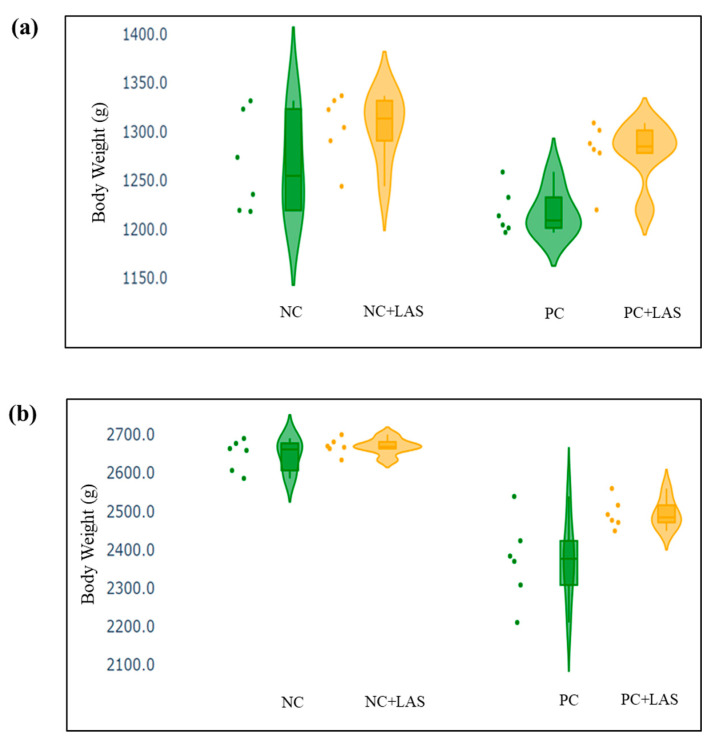
(**a**) Scatter and violin plot of body weight (BW) measured at 24 d of age. (**b**) Scatter and violin plot of BW measured at 42 d of age.

**Figure 5 toxins-17-00212-f005:**
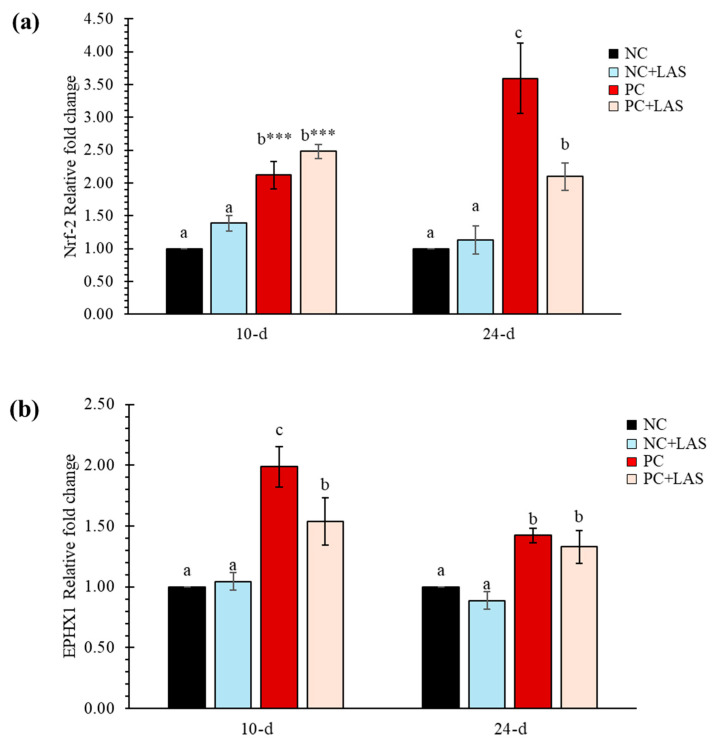
(**a**) The relative mRNA expression (2^−ΔΔCt^) of the Nrf-2 gene in hepatic tissues. The means represent 6 birds per treatment (n = 1 bird/replicate); bars with dissimilar superscripts in a row varied significantly; ANOVA P indicated by *** at *p* < 0.001. For 2^−ΔΔCt^-value of Nrf-2 gene: Main effect contamination: At 10 d *p* = 0.011, At 24 d *p* = 0.038; main effect LAS: At 10 d *p* = 0.001, At 24 d *p* = 0.001; contamination *LAS interaction: At 10 d *p* = 0.928, At 24 d *p* = 0.016. (**b**) The relative mRNA expression (2^−ΔΔCt^) of the EPHX1 gene in hepatic tissues. For the 2^−ΔΔCt^-value of the EPHX1 gene: main effect contamination: At 10 d *p* = 0.144, At 24 d *p* = 0.219; main effect LAS: At 10 d *p* = 0.001, At 24 d *p* = 0.001; contamination*LAS interaction: At 10 d *p* = 0.078, At 24 d *p* = 0.915.

**Table 1 toxins-17-00212-t001:** The chemical composition of the experimental diets (analyzed values *).

Nutrients (g/100 g as Fed)	Standard Diet		Contaminated Diet	
	Without LAS	With LAS	Without LAS	With LAS
Starter (1–10 d)				
Dry matter	88.75	89.02	88.95	89.10
Organic matter	95.75	95.26	95.14	95.28
Crude protein	23.27	23.38	23.44	23.29
Ether extract	3.61	3.38	3.49	3.58
Crude fiber	3.91	3.95	3.89	3.87
Calcium	0.885	0.879	0.865	0.881
Total phosphorus	0.628	0.641	0.639	0.655
Aflatoxins μg/kg **	2.3	-	76.2	-
Fumonisins μg/kg **	716.6	-	1602.2	-
Ochratoxin A μg/kg **	6.2	-	57.8	-
Grower (11–24 d)				
Dry matter	88.82	88.8	88.84	88.78
Organic matter	95.45	95.67	95.53	95.55
Crude protein	20.54	20.62	20.58	20.37
Ether extract	4.15	4.11	4.05	4.11
Crude fibre	2.85	2.88	2.79	2.68
Calcium	0.757	0.748	0.777	0.755
Total Phosphorus	0.576	0.542	0.557	0.542
Aflatoxins μg/kg **	3.2	-	83.1	-
Fumonisins μg/kg **	1000.2	-	1600.2	-
Ochratoxin A μg/kg **	12.2	-	54.3	-
Finisher (25–42 d)				
Dry matter	89.05	88.92	89.09	88.88
Organic matter	95.99	96.25	96.18	96.37
Crude protein	19.68	19.75	19.56	19.76
Ether extract	5.78	5.82	5.75	5.66
Crude fiber	2.57	2.39	2.38	2.25
Calcium	0.657	0.628	0.644	0.628
Total Phosphorus	0.505	0.522	0.508	0.505
Aflatoxins μg/kg **	4.2	-	79.4	-
Fumonisins μg/kg **	850.3	-	1702.1	-
Ochratoxin A μg/kg **	10.2	-	54.2	-

* Analyzed values dry matter (DM, method number 934.01), nitrogen and crude protein (CP, method number 954.01), ether extract (EE, method number 920.39), crude fiber (CF, method number 962.09/AOAC 978.10), total ash (AOAC 942.05-2012), organic matter (OM, method number 942.05), and calcium and total phosphorus (AOAC 927.02-1990 and AOAC 965.17-1966). ** The “without LAS” and the “with LAS” groups under the standard and the contaminated diets were the same. Hence, toxin analysis was performed in each one of these diets.

**Table 2 toxins-17-00212-t002:** The effects of supplementation of the liquid anti-mycotoxin solution (LAS) through drinking water for 1–42 d on the performance of male broiler chickens fed a diet naturally contaminated with mycotoxins ^1^.

Attributes	Standard Diet		Contaminated Diet		Pooled	Main Effects-P		Interaction-P
	Without LAS	With LAS	Without LAS	With LAS	SEM	Contamination	LAS	Contamination*LAS
Body Weight (g)								
0 d	44.5	44.8	44.5	44.7	0.15	0.755	0.105	0.755
10 d	318.9 ^ab^	326.8 ^ab^	315.4 ^a^	330.5 ^b^	6.96 *	0.984	0.003	0.298
24 d	1267.2 ^ab^	1305.3 ^b^	1218.1 ^a^	1279.9 ^b^	36.62 **	0.022	0.003	0.437
42 d	2647.1 ^c^	2669.2 ^c^	2372.7 ^a^	2494.3 ^b^	139.2 *	<0.0001	0.011	0.067
Average Daily Gain (g)								
1–10 d	27.43 ^ab^	28.21 ^ab^	27.09 ^a^	28.59 ^b^	0.69 *	0.961	0.003	0.297
11–24 d	67.74 ^ab^	69.89 ^b^	64.48 ^a^	67.81 ^ab^	2.24 **	0.014	0.012	0.558
25–42 d	76.66 ^b^	75.78 ^b^	64.14 ^a^	67.47 ^a^	6.17 *	<0.0001	0.435	0.184
1–42 d	61.97 ^c^	62.49 ^c^	55.43 ^a^	58.33 ^b^	3.31 *	<0.0001	0.011	0.067
Feed Intake (g)								
1–10 d	296.9 ^ab^	306.2 ^b^	296.6 ^a^	311.8 ^c^	7.43 *	0.459	0.002	0.41
11–24 d	1238.7 ^b^	1263.8 ^c^	1223.7 ^a^	1262.2 ^bc^	19.37	0.611	0.062	0.679
25–42 d	2524.8 ^c^	2472.1 ^bc^	2348.9 ^b^	2290.2 ^a^	108.16 **	0.001	0.253	0.95
1–42 d	4060.4 ^bc^	4042.1 ^c^	3869.2 ^b^	3864.3 ^a^	106.79 **	0.0005	0.797	0.883
Average Daily Feed Intake (g)								
1–10 d	29.69 ^a^	30.62 ^ab^	29.66 ^a^	31.18 ^b^	0.74 *	0.459	0.002	0.41
11–24 d	88.48 ^b^	90.27 ^c^	87.41 ^a^	90.16 ^bc^	1.38	0.611	0.062	0.679
25–42 d	140.26 ^ab^	137.34 ^ab^	130.5 ^a^	127.23 ^a^	6.01 **	0.001	0.253	0.95
1–42 d	96.68 ^b^	96.24 ^b^	92.12 ^a^	92.01 ^a^	2.54 **	0.001	0.797	0.883
Feed Conversion Ratio								
1–10 d	1.083	1.086	1.095	1.091	0.003	0.103	0.915	0.511
11–24 d	1.307 ^ab^	1.292 ^a^	1.356 ^b^	1.33 ^ab^	0.009 *	0.008	0.185	0.719
25–42 d	1.831 ^a^	1.814 ^a^	2.04 ^b^	1.885 ^a^	0.024 ***	0.001	0.014	0.045
1–42 d	1.56 ^a^	1.54 ^a^	1.663 ^b^	1.577 ^a^	0.012 ***	<0.0001	0.001	0.02
Liveability (%)	95.8	98.6	93.1	97.2	2.37	0.359	0.133	0.757
EPI	387.2 ^b^	407.1 ^c^	317.4 ^a^	366.1 ^b^	38.52 ***	<0.0001	0.006	0.206

^1^ The means represent 6 birds per treatment (n = 1 bird/replicate); means with dissimilar superscripts in a row varied significantly; ANOVA P indicated by * at *p* < 0.05, ** *p* < 0.01, and *** *p* < 0.001.

**Table 3 toxins-17-00212-t003:** The effects of supplementation using the liquid anti-mycotoxin solution (LAS) through drinking water for 1–42 d on the gross carcass traits and relative weight of the internal organs of male broiler chickens fed with a diet naturally contaminated with mycotoxins ^1^.

Attributes	Non-Contaminated Feed		Contaminated Feed		Pooled SEM	Main Effect-P		Interaction-P
	Without LAS	With LAS	Without LAS	With LAS		Contamination	LAS	Contamination*LAS
**Absolute** **weights g**								
Dressed carcass weight	1862.2	1898.8	1874.0	1859.5	17.91	0.729	0.78	0.52
Liver	57.17	52.5	52.17	53.92	2.28	0.57	0.644	0.314
Heart	8.25	8.5	9.67	8.5	0.64	0.309	0.507	0.309
Gizzard	22.67	23.17	23.75	24.92	0.96	0.355	0.583	0.826
Giblets	88.08	84.17	85.58	87.33	1.76	0.932	0.781	0.469
Abdominal fat	42.33	39.75	42.42	45.83	2.49	0.501	0.927	0.513
Breast	618.92	661.5	621.58	617.75	21.1	0.239	0.266	0.186
Drumstick	217.5 ^ab^	227.0 ^b^	214.4 ^a^	234.7 ^c^	9.26 *	0.671	0.012	0.329
Thighs	254.9	258.5	262.4	271.8	7.27	0.203	0.416	0.719
**Relative weight g/kg live weight**								
Dressing yield	748.3	758.9	742.7	746.3	6.93	0.269	0.386	0.669
Liver	23.01 ^c^	20.99 ^ab^	20.7 ^a^	21.65 ^b^	1.02 *	0.522	0.679	0.257
Heart	3.32 ^a^	3.41 ^b^	3.86 ^c^	3.42 ^b^	0.24 *	0.371	0.558	0.379
Gizzard	9.13 ^a^	9.28 ^ab^	9.43 ^b^	10.02 ^c^	0.38 **	0.431	0.567	0.732
Giblets	35.47 ^c^	33.69 ^a^	33.99 ^ab^	35.09 ^b^	0.85 *	0.98	0.84	0.397
Abdominal fat	16.97 ^b^	15.78 ^a^	16.73 ^ab^	18.43 ^c^	1.09 *	0.481	0.881	0.398
Breast	248.7 ^b^	264.67 ^c^	246.52 ^a^	247.79 ^ab^	8.55 *	0.104	0.139	0.204
Drumstick	87.44 ^ab^	90.75 ^b^	85.03 ^a^	94.29 ^c^	4.03 *	0.763	0.003	0.128
Thighs	102.42 ^a^	103.33 ^ab^	103.92 ^b^	109.1 ^c^	3.02 **	0.123	0.192	0.355

^1^ The means represent 6 birds per treatment (n = 1 bird/replicate); means with dissimilar superscripts in a row varied significantly; ANOVA P indicated by * at *p* < 0.05 and ** *p* < 0.01.

**Table 4 toxins-17-00212-t004:** The effects of supplementation using the liquid anti-mycotoxin solution (LAS) through drinking water for 1–42 d on different liver function parameters of male broiler chickens fed with a diet naturally contaminated with mycotoxins ^1^.

Attributes	Non-Contaminated Feed		Contaminated Feed		Pooled SEM	Main Effect-P		Interaction-P
	Without LAS	With LAS	Without LAS	With LAS		Contamination	LAS	Contamination*LAS
GSH-Px (u/L)								
10 d	1.99	2.12	1.92	2.26	0.099	0.87	0.254	0.61
24 d	1.79	2.03	1.87	2.20	0.084	0.458	0.104	0.806
SOD (u/L)								
10 d	61.72	52.64	64.36	63.40	2.58	0.205	0.338	0.438
24 d	92.41 ^c^	56.98 ^a^	77.07 ^b^	113.28 ^d^	4.74 ***	0.001	0.928	0.001
ALP (μKat/L)								
10 d	1408.1	1778.6	2198.9	1652.0	181.4	0.377	0.813	0.226
24 d	628.3	755.0	838.4	836.6	54.07	0.198	0.574	0.564
AST (μKat/L)								
10 d	2.99 ^a^	3.34 ^ab^	3.53 ^ab^	3.65 ^b^	0.1 *	0.034	0.22	0.52
24 d	3.87	4.80	3.83	4.30	0.25	0.594	0.174	0.644
ALT (μKat/L)								
10 d	0.114	0.113	0.118	0.114	0.003	0.657	0.657	0.781
24 d	0.109	0.126	0.114	0.126	0.004	0.777	0.12	0.763
γ-GT (μKat/L)								
10 d	0.149	0.142	0.157	0.161	0.004	0.076	0.853	0.481
24 d	0.163 ^a^	0.223 ^ab^	0.263 ^b^	0.244 ^b^	0.013 *	0.01	0.349	0.077

^1^ The means represent 6 birds per treatment (n = 1 bird/replicate); means with dissimilar superscripts in a row varied significantly; ANOVA P indicated by * at *p* < 0.05 and *** *p* < 0.001. Age effect GSH-Px (reduced glutathione) *p* = 0.432; age*treatment *p* = 0.97. Age effect SOD (superoxide dismutase) *p* < 0.001; age*treatment *p* < 0.001. Age effect ALP (alkaline phosphatase) *p* < 0.001; age*treatment *p* = 0.708. Age effect AST (aspartate amino transferase) *p* = 0.005; age*treatment *p* = 0.476. Age effect ALT (alanine amino transferase) *p* < 0.495; age*treatment *p* = 0.495. Age effect γ-GT (γ-glutamyl transferase) *p* < 0.001; age*treatment *p* = 0.025

**Table 5 toxins-17-00212-t005:** Chemical composition of experimental diets (analyzed values *).

Ingredients (g/kg)	Standard Diet			Contaminated Diet		
	Starter	Grower	Finisher	Starter	Grower	Finisher
Maize–standard	588.0	642.1	673.47	187.0	241.1	273.47
Maize–contaminated ^1^	-	-	-	400.0	400.0	400.0
Soybean meal ^2^	220.6	174.3	147.0	175.6	129.3	102.0
Full fat soybean	60.0	60.0	60.0	60.0	60.0	60.0
Maize gluten meal–standard	40.0	40.0	40.0	-	-	-
Maize gluten meal–contaminated ^3^	-	-	-	55.0	55.0	55.0
Groundnut cake ^4^	-	-	-	70.0	70.0	70.0
De-Oiled rape Seed meal	40.0	40.0	40.0	-	-	-
Soybean oil	8.4	9.4	10.8	8.4	9.4	10.8
Mono calcium phosphate	16.9	11.8	7.87	16.9	11.8	7.64
Limestone powder	8.1	6.0	5.98	8.0	6.1	5.58
DL-Methionine	3.2	2.8	2.41	3.2	2.8	2.41
L-Lysine HCl	4.0	3.7	3.29	4.3	4.0	3.5
L-Threonine	1.5	1.2	0.86	1.8	1.5	1.0
L-Arginine	1.0	1.05	0.82	1.3	1.2	1.0
L-Valine	0.5	0.45	0.4	0.7	0.6	0.5
Salt	2.4	2.1	2.0	2.4	2.1	2.0
Sodium-bi-carbonate	2.3	2.0	2.0	2.3	2.0	2.0
Vitamin premix ^5^	1.0	1.0	1.0	1.0	1.0	1.0
Trace minerals ^6^	0.5	0.5	0.5	0.5	0.5	0.5
Salinomycin 12%	0.5	0.5	0.5	0.5	0.5	0.5
Phytase ^7^	0.1	0.1	0.1	0.1	0.1	0.1
Choline chloride 60%	1.0	1.0	1.0	1.0	1.0	1.0
AME kcal/kg	2975	3050	3100	2975	3050	3100
Crude protein	22.27	20.100	19.000	22.27	20.100	19.000
Calcium	0.857	0.674	0.604	0.857	0.674	0.604
Available phosphorus	0.50	0.42	0.36	0.50	0.42	0.36
SID Amino acids						
Lysine	1.32	1.18	1.08	1.32	1.18	1.08
Methionine	0.655	0.593	0.544	0.655	0.593	0.544
Met + Cys	1.05	0.92	0.90	1.05	0.92	0.90
Threonine	0.88	0.79	0.72	0.88	0.79	0.72
Tryptophan	0.24	0.21	0.2	0.24	0.21	0.2
Arginine	1.4	1.27	1.27	1.4	1.27	1.27
Isoleucine	0.88	0.8	0.753	0.88	0.8	0.753
Valine	0.96	0.91	0.84	0.96	0.91	0.84
Leucine	1.83	1.718	1.65	1.83	1.718	1.65
Histidine	0.52	0.47	0.443	0.52	0.47	0.443
Phenylalanine	1.0	0.918	0.867	1.0	0.918	0.867
Glycine	0.56	0.534	0.424	0.56	0.534	0.424
Serine	0.7	0.627	0.49	0.7	0.627	0.49
Sodium	0.22	0.2	0.2	0.22	0.2	0.2
Chloride	0.2	0.18	0.18	0.2	0.18	0.18
Potassium	0.88	0.85	0.754	0.88	0.85	0.754
Linoleic acid	2.0	>2	>2	2.0	>2	>2
Choline mg/kg	1700	1600	1500	1700	1600	1500

^1^ Contains ≥ 50 ppb aflatoxins and ≥6000 ppb fumonisins; ^2^ analyzed to contain 48% crude protein; ^3^ contains ≥ 150 ppb aflatoxins and ≥100 ppb ochratoxin; ^4^ analyzed to contain 100 ppb aflatoxins; ^5^ contains (per kg) vitamin A 13.5 MIU, vitamin D3 4.5 MIU, vitamin E 60 g, vitamin K3 3.5 g, vitamin B1 3.5 g, vitamin B2 8.0 g, vitamin B6 3.5 g, vitamin B12 0.02 g, biotin 0.145 g, pantothenic acid 14.5 g, folic acid 2.25 g, and niacin 60 g) ^6^ contains (g/kg) manganese (100), zinc (80), copper (15), and iron (90) in the form of sulfate salts, and iodine (2.0) as potassium iodide; ^7^ *Buttiauxella* phytase has a declared phytase activity of 10,000 FTU/g.

**Table 6 toxins-17-00212-t006:** The sequence of the primers used in the mRNA expression analysis.

Oligo	Forward Primer (5′-3′)	Reverse Primer (3′-5′)	Size (bp)	Accession
NRF-2	AGAAAACGCTGAACCACCAATC	GCTGGGTGGCTGAGTTTGATTA	217	NM_205117.1
EPHX1	GAAGATGTCAGGCGGATGTT	CAGGAGAGTCATTCAAACCACA	127	XM_419386.5
GAPDH	GCAGATGCAGGTGCTGAGTA	GACACCCATCACAAACATGG	144	NM_204305.1

## Data Availability

The original contributions presented in this study are included in the article. Further inquiries can be directed to the corresponding author.

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
