# Peer review of "The Efficacy of a Novel Water-Soluble Anti-Mycotoxin Solution in Improving Broiler Chicken Performance Under Mycotoxin Challenge"

_toxins, 2025, doi:10.3390/toxins17050212_

Round 1
Reviewer 1 Report
Comments and Suggestions for Authors
The present article is super interesting and informative for the scientific community. However, very minor errors were detected.
Abstract
- Why was the liver selected for oxidative defense genes and the spleen for inflammatory cytokines?
- Line 15- Explain the changes observed in the oxidative defense genes and inflammatory cytokines.
- Line 17-19- The authors stated that “LAS supplementation improved BW and FCR, reduced Nrf-2 expression, and enhanced mycotoxin detoxification via lower EPHX1 expression”. However, it is well-established that the Nrf2 signaling pathway plays an essential role in mitigating oxidative stress by upregulating antioxidant genes. Typically, an increase in Nrf2 expression is associated with improved detoxification of harmful agents, including mycotoxins. The reported reduction in Nrf2 expression seems contradictory, as lower Nrf2 activity might weaken the antioxidant response. Additionally, numerous studies have shown that plant herbs supplementation can enhance Nrf2 expression, potentially providing a protective effect against oxidative stress induced by mycotoxins. The authors may need to clarify or further explain the mechanistic basis for this observation.
Results
- How do the results confirm the scientific hypothesis presented in the manuscript title? The author should systematically analyze and summarize the results based on this logic.
Conclusion
- Delete bullet points and re-write the conclusion section in paragraphs.
Material and Methods
How did the author select the doses for this experiment? Any references or pilot study?
Author Response
Abstract
Comment 1: Why was the liver selected for oxidative defense genes and the spleen for inflammatory cytokines?
Response 1: Thank you for the observation. Liver is the primary organ responsible for detoxification and metabolism of mycotoxins. Mycotoxins like aflatoxins induce oxidative stress in the liver by generating reactive oxygen species, thus disrupting the cellular function. Genes such as Nrf-2 and EPHX1 are crucial in oxidative stress response and detoxification, making the liver the ideal organ for studying oxidative defense mechanisms.
Regarding the spleen, it was mistakenly mentioned in the abstract, as it was not included in the study. We have corrected the abstract as follows:
[Mycotoxins like aflatoxins (AFs), fumonisins (FBs), and ochratoxin A (OTA) poses a serious health risk to humans and animals. Fruit pomace extracts, rich in natural nutrients and bioactive compounds have the potential in enhancing animal health and mitigating mycotoxin toxicity. This study evaluated a novel liquid anti-mycotoxin solution (LAS), a combination of grape and olive pomace extract administered through drinking water (2L:1000L) from 1-42 days in broiler chickens under a natural multi-mycotoxin challenge. The 42-day trial with 288, one-day-old male Ross 308AP95 included four experimental groups: a negative control (NC), NC+LAS, a positive control (PC) group fed a diet containing 80 μg/kg AFs, 1600 μg/kg FBs, and 50 μg/kg OTA and PC+LAS. Growth performance, oxidative defense genes (liver), and stress biomarkers (blood) were analyzed. Mycotoxin exposure negatively affected body weight (BW), feed conversion ratio (FCR) and oxidative defense mechanism. LAS supplementation improved BW and FCR, reduced Nrf-2 expression, and enhanced mycotoxin detoxification via lower EPHX1 expression. Though LAS did not fully restore performance to NC levels, it significantly mitigated mycotoxin-induced damage. This study concluded that LAS can be a promising solution to improve broiler resilience against moderate to high mycotoxin exposure.]
Comment 2: Line 15- Explain the changes observed in the oxidative defense genes and inflammatory cytokines.
Response 2: Thank you for your query. Here, I would like to clarify that expression of inflammatory cytokines was beyond the scope of the present study, hence no observations were recorded.
Regarding oxidative defense genes it is mentioned in the abstract that LAS supplementation improved BW and FCR, reduced Nrf-2 expression, and enhanced mycotoxin detoxification via lower EPHX1 expression. The specific fold changes are detailed in the results section but were avoided in the abstract due to word limitations.
Comment 3: Line 17-19- The authors stated that “LAS supplementation improved BW and FCR, reduced Nrf-2 expression, and enhanced mycotoxin detoxification via lower EPHX1 expression”. However, it is well-established that the Nrf2 signaling pathway plays an essential role in mitigating oxidative stress by upregulating antioxidant genes. Typically, an increase in Nrf2 expression is associated with improved detoxification of harmful agents, including mycotoxins. The reported reduction in Nrf2 expression seems contradictory, as lower Nrf2 activity might weaken the antioxidant response. Additionally, numerous studies have shown that plant herbs supplementation can enhance Nrf2 expression, potentially providing a protective effect against oxidative stress induced by mycotoxins. The authors may need to clarify or further explain the mechanistic basis for this observation.
Response 3: We acknowledge that Nrf2 plays a crucial role in mitigating oxidative stress by upregulating antioxidant defense mechanisms. Typically, an increase in Nrf2 expression is associated with enhanced detoxification and protection against oxidative damage. However, our observation of reduced Nrf2 expression in LAS-supplemented birds requires careful interpretation. LAS supplementation significantly mitigated mycotoxin-induced oxidative stress, which may have reduced the need for an exaggerated Nrf2 response. As oxidative stress levels decreased, a negative feedback mechanism could have downregulated Nrf2 expression, as the activation of this pathway is often stress-dependent.
Results
Comment 4: How do the results confirm the scientific hypothesis presented in the manuscript title? The author should systematically analyze and summarize the results based on this logic.
Response 4: Thank you for your query. The study aimed to evaluate the efficacy of a novel water-soluble anti-mycotoxin solution (LAS) in improving broiler chicken performance under mycotoxin challenge. The results support the hypothesis in the following ways:
Growth Performance: Mycotoxin contamination significantly reduced BW, ADG, and FCR, confirming the expected negative impact of mycotoxins. However, LAS supplementation improved BW and FCR, particularly during later growth stages (P < 0.05), indicating its efficacy in mitigating mycotoxin-induced performance losses.
Hepatic and Antioxidant Biomarkers: Mycotoxin exposure induced mild to moderate changes in hepatic enzyme activities (AST, γ-GT), but the overall impact was not severe. LAS supplementation influenced SOD activity, with differential effects in birds fed the standard vs. contaminated diet (P < 0.001). While LAS did not significantly alter other biomarkers, the observed trends suggest a potential role in mitigating oxidative stress under mycotoxin challenge.
Oxidative Defense Gene Expression: Nrf-2 expression was upregulated in response to mycotoxin exposure, confirming oxidative stress induction. LAS supplementation downregulated Nrf-2 expression at 24 days (P = 0.001), indicating its role in oxidative stress mitigation. Similarly, EPHX1 expression was lower in the PC+LAS group at 10 days compared to the PC group, suggesting improved detoxification capacity.
Hence, the results confirm the hypothesis that LAS supplementation mitigates the negative effects of mycotoxins on broiler performance by improving BW and FCR, modulating oxidative stress responses (Nrf-2, EPHX1), and influencing antioxidant activity (SOD).
Conclusion
Comment 5: Delete bullet points and re-write the conclusion section in paragraphs.
Response 5: The present study established that feeding a mycotoxin-contaminated diet with significantly high concentrations of AFs, FBs, and OTA negatively impacted BW and FCR in broiler chickens. However, supplementation with the LAS resulted in an economically considerable improvement in BW and FCR, particularly in birds exposed to mycotoxin contaminated diet. While growth performance was affected by mycotoxin exposure, gross carcass traits and internal organ weights relative to live weight remained unaffected, regardless of diet composition or LAS supplementation.
In regards to oxidative stress response, LAS supplementation decreased superoxide dismutase activity in birds fed with the non-contaminated diet but increased in mycotoxins contaminated diet. The study also indicated that feeding the contaminated diet negatively affected the oxidative defense mechanism, as evidenced by the upregulated expression of Nrf-2. LAS supplementation appeared to mitigate this effect by reducing Nrf-2 expression. Similarly, the expression of EPHX1, which plays a role in the detoxification of mycotoxins, was comparatively lower in LAS-supplemented birds, suggesting a protective effect of LAS against mycotoxin-induced stress.
Overall, it was concluded from the present study that supplementation of the liquid anti-mycotoxin solution (LAS) at a concentration of 2L:1000L from 1-42 d in broiler chickens was capable of alleviating the negative effects of a natural multi-contaminated diet by high levels of AFs, OTA and moderate levels of FBs. Although, the BW and FCR of the supplemented birds fed with the contaminated diet were not comparable with those of the birds fed with the standard diet, a substantial improvement may be possible with LAS supplementation in cases of impending moderate to high levels of mycotoxin challenge.
Material and Methods
Comment 6: How did the author select the doses for this experiment? Any references or pilot study?
Response 6: We really appreciate the reviewer’s comment. The dosage of the LAS such as the inclusion of the natural extracts in the product formulation, was selected according to the literature available regarding the grape pomace and olive pomace extracts in animal nutrition, which was previously presented in Toxins journal in the review regarding Promising Phytogenic Feed Additives Used as Anti-Mycotoxin Solutions in Animal Nutrition by Quesada et al. (2024). A summary table collecting all references is shown in the supplementary data file of the article aforementioned. In addition, the company performed in vitro tests to assess the efficacy and safety of the formulation that have not been published in a scientific journal yet. With the information collected, the LAS dosage recommendations were established within 0.5 L to 2 L for 1000 L of drinking water in poultry.
Reference:
Quesada-Vázquez, S.; Codina Moreno, R.; Della Badia, A.; Castro, O.; Riahi, I. Promising Phytogenic Feed Additives Used as Anti-Mycotoxin Solutions in Animal Nutrition. Toxins 2024, 16, 434. https://doi.org/10.3390/toxins16100434
Reviewer 2 Report
Comments and Suggestions for Authors
In this study, the authors evaluated the efficacy of a liquid fruit pomace extract in mitigating the toxicological effect of AFs, Fbs and OTA in broiler chickens. The results showed that this fruit pomace extract improved resilience against mycotoxin exposure and restored growth performance of broiler chickens. In a nutshell, this study is evidentally solid and of significance to poultry production, and I suggest this manuscript to be accepted after minor revision.
1. Line 66, besides the mentioned products, there are also biodegradation bacterial preparations for the detoxification of mycotoxin, such as BBSH 797 from Biomin.
2. Sector 3 Discussion, could the authors give a brief review about the main chemical components in the LAS?
Author Response
Comment 1. Line 66, besides the mentioned products, there are also biodegradation bacterial preparations for the detoxification of mycotoxin, such as BBSH 797 from Biomin.
Response 1: Thank you for the suggestion. It is added in line 86-87.
In response to this challenge, feed additives such as mycotoxin adsorbents or binders, enzymes, probiotics, prebiotics, bacterial preparations, phytogenics and antioxidants, have emerged as a practical strategy to mitigate the harmful effects of mycotoxins in animal nutrition. These additives work by binding mycotoxins in the gastrointestinal tract to reduce their bioavailability, by enzymatically degrading them into less harmful compounds or by boosting the systemic health with natural extracts rich in phytogenics with antioxidant, anti-inflammatory and hepatoprotective properties to mitigate the detrimental effects of mycotoxins [19, 20].
Comment 2: Sector 3 Discussion, could the authors give a brief review about the main chemical components in the LAS?
Response 2: LAS grape pomace extract is rich in polyphenolic compounds such as proanthocyanidins, anthocyanins, procyanidins, catechin, ferulic acid, quercetin, resveratrol and syringic acid, among other bioactive compounds and derivates.
LAS olive pomace extract is rich in hydroxytyrosol, tyrosol and other polyphenolic derivates.
Besides, LAS contains citric acid, sorbic acid, water and sorbitol of high purity as chemical components.
Reviewer 3 Report
Comments and Suggestions for Authors
Dear Authors,
This is an excellent and timely choice of topic, which could have huge economic and food safety implications..
- This study primarily focused on aflatoxins (AFs), fumonisins (FBs), and ochratoxin A (OTA). While these are important mycotoxins, many others can affect poultry. The effectiveness of LAS against a broader range of mycotoxins has not yet been addressed.
- Single dosage testing: The study only evaluated LAS at one concentration (2 L: 1000 L). It is unclear whether this is the optimal dosage or whether different concentrations might be more effective.
- The study was conducted over 42 days, which may not be sufficient to assess the long-term effects or potential toxicity of prolonged LAS use.
- Although LAS improved performance, it did not fully restore it to the levels of the negative control group. This suggests that the solution may have limitations in terms of efficacy.
- Why not measure the possible mycotoxin content of the LAS? Why is there no description of the LAS in the Materials and Methods?
- This study did not address the economic feasibility of using LAS in commercial poultry production. Compared to other mycotoxin binders, how much more expensive is this „new product”? What was the meat quality of the animals?
- This study only used male Ross 308AP95 chicks, which may limit the generalisability of the results to other breeds or female chickens.
- Potential confounding factors: The study did not discuss potential interactions between LAS and other feed components or environmental factors that could influence the results.
- Although some gene expression data were provided, the study lacked a comprehensive explanation of the molecular mechanisms by which LAS exerts its effects.
Congratulations on your work so far; keep up this line of research; it is a great opportunity!
Author Response
Comment 1: This study primarily focused on aflatoxins (AFs), fumonisins (FBs), and ochratoxin A (OTA). While these are important mycotoxins, many others can affect poultry. The effectiveness of LAS against a broader range of mycotoxins has not yet been addressed.
Response 1: We appreciate the reviewer’s comment. In this study, we selected aflatoxins (AFs), fumonisins (FBs), and ochratoxin A (OTA) as the primary mycotoxins for evaluation, as these are among the most prevalent and toxic contaminants routinely detected in poultry feed worldwide. Their co-occurrence is frequently reported and known to synergistically impact poultry health and performance. While this study demonstrates the potential of LAS in mitigating the effects of these key mycotoxins, we acknowledge the importance of assessing the solution’s efficacy against other relevant mycotoxins (e.g., zearalenone, T-2 toxin, and deoxynivalenol) and plan to address this in future research.
Comment 2: Single dosage testing: The study only evaluated LAS at one concentration (2 L: 1000 L). It is unclear whether this is the optimal dosage or whether different concentrations might be more effective.
Response 2: We really appreciate the reviewer’s comment. The dosage of the LAS such as the inclusion of the natural extracts in the product formulation, was selected according to the literature available regarding the grape pomace and olive pomace extracts in animal nutrition, which was previously presented in Toxins journal in the review regarding Promising Phytogenic Feed Additives Used as Anti-Mycotoxin Solutions in Animal Nutrition by Quesada et al. (2024). A summary table collecting all references is shown in the supplementary data file of the article aforementioned. In addition, the company performed in vitro tests to assess the efficacy and safety of the formulation that have not been published in a scientific journal yet. With the information collected, the LAS dosage recommendations were established within 0.5 L to 2 L for 1000 L of drinking water in poultry. Since the study presented in this publication corresponds to the first scientific trial of the LAS in vivo in broilers, only the maximum recommended dosage was tested in order to facilitate observing the effects of the product administered through drinking water. However, after proving that LAS has beneficial effects on broilers, we agree that in future studies other doses should be included in the experimental design to assess the efficacy of the LAS recommendations range.
Reference:
Quesada-Vázquez, S.; Codina Moreno, R.; Della Badia, A.; Castro, O.; Riahi, I. Promising Phytogenic Feed Additives Used as Anti-Mycotoxin Solutions in Animal Nutrition. Toxins 2024, 16, 434. https://doi.org/10.3390/toxins16100434
Comment 3: The study was conducted over 42 days, which may not be sufficient to assess the long-term effects or potential toxicity of prolonged LAS use.
Response 3: Thank you for this observation. The current study was specifically designed for commercial broiler production, where birds are typically marketed or processed by 42 days of age. Therefore, evaluating effects beyond this period was beyond the scope of the present trial. We agree, however, that assessing the long-term safety and efficacy of LAS in longer-living poultry categories, such as breeders or layers, is important. We intend to conduct separate, targeted studies in these populations to evaluate chronic exposure and any potential cumulative effects of LAS use.
Comment 4: Although LAS improved performance, it did not fully restore it to the levels of the negative control group. This suggests that the solution may have limitations in terms of efficacy.
Response 4: It is but obvious that when an infection/challenge is imparted the birds suffer a substantial amount of nutrient loss in combatting that challenge. Mycotoxin challenge is not an exception. In this study the challenge was moderate in nature and the effect was chronic and continuous. This means the birds were under continuous exposure. Moreover, LAS was introduced through drinking water and therefor it is quite possible that the binder/neutralizer and the toxins entered simultaneously into the gut and some toxins/metabolites might escape into the systemic circulation through the window between the intake and release of the toxin from diet and the actual contact happening between the toxin and LAS. This might be one of the contributory factors that 100% restoration of performance was not achieved.
Comment 5: Why not measure the possible mycotoxin content of the LAS? Why is there no description of the LAS in the Materials and Methods?
Response 5: Thank you for your comment. The mycotoxin content of the LAS was not measured in this study; however, the same batch of LAS was used for both treatments that included the product meaning that if there was any mycotoxin contamination provided by LAS was the same for both treatments. In addition, we added a section in materials and methods describing the LAS composition as follows:
The liquid anti-mycotoxin solution (LAS) used in this study is a multi-component liquid mycotoxin-detoxifying solution administered through drinking water. This solution contains natural phytogenic compounds derived from grape (Vitis Vinifera) and olive (Olea Europaea) pomace extracts, that are rich in a wide variety of bioactive polyphenolic compounds such as phenolic acids, proanthocyanidins, anthocyanins and flavonoids. Additionally, the formulation includes organic acids (citric acid, sorbic acid), which contribute to the improvement of the water quality and intestinal health due to their antimicrobial properties. Furthermore, it contains sorbitol, an emulsifying agent that enhances the stability and oral bioavailability of the phytogenic molecules. Therefore, the beneficial effects of the ingredients included in the liquid product formulation contribute to mitigate the detrimental effects of mycotoxins exposure.
Comment 6: This study did not address the economic feasibility of using LAS in commercial poultry production. Compared to other mycotoxin binders, how much more expensive is this “new product”? What was the meat quality of the animals?
Response 6: Thank you for your observation. The LAS is aimed to complement regular mycotoxin binders added through the diet in challenging scenarios, such as stress induced by high levels of mycotoxin contamination, heat stress, or other stressful conditions, and to support the overall health of the birds. Therefore, LAS is not comparable to regular mycotoxin binders for feed; it’s a different category of product, so its costs are not comparable either. LAS costs are comparable to other premium solutions, including natural extracts with antioxidant, hepatoprotective and detoxifying properties, that enhance the systemic health of the animals and are administered through drinking water.
Regarding the meat quality of the animals, we would like to add that meat quality assessment was not under the scope of this study and hence it was not assessed.
Comment 7: This study only used male Ross 308AP95 chicks, which may limit the generalisability of the results to other breeds or female chickens.
Response 7: We appreciate the reviewer’s observation. Ross and Vencobb are two of the most commonly used commercial broiler strains, with similar nutritional requirements, growth patterns, and responses to dietary interventions. Therefore, using male Ross 308AP95 birds offers a practical and representative model for commercial broiler production. Additionally, using a single strain and sex helps reduce biological variability and allows for clearer interpretation of treatment effects, particularly in mechanistic and performance studies. While we believe the current findings are broadly applicable, we acknowledge that further studies involving other strains and female birds would help confirm the generalizability, and such investigations can be planned in future research.
Comment 8: Potential confounding factors: The study did not discuss potential interactions between LAS and other feed components or environmental factors that could influence the results.
Response 8: We appreciate the reviewer’s concern regarding potential confounding factors. This study was designed to closely mimic practical field conditions under a controlled facility, focusing on a naturally contaminated feed matrix containing three major mycotoxins commonly found in poultry feed where poultry production is intensive.
The primary objective of the study was to evaluate the efficacy of the LAS against the detrimental effects of the common mycotoxins. While we acknowledge that environmental and dietary factors (e.g., feed composition, housing temperature, or humidity) may potentially modulate bird responses, every effort was made to maintain uniform housing, feeding, and management conditions across all experimental groups to minimize external variability. Additionally, the feed formulation was kept consistent across treatments, except for the presence or absence of LAS and the natural mycotoxin challenge. No other feed additives or environmental interventions were introduced that could confound the evaluation of LAS efficacy.
Comment 9: Although some gene expression data were provided, the study lacked a comprehensive explanation of the molecular mechanisms by which LAS exerts its effects.
Response 9: We appreciate the reviewer’s insightful observation. As correctly noted, the current study included gene expression analysis focusing on Nrf-2 and EPHX1, which are pivotal markers of oxidative stress response and xenobiotic detoxification, respectively. These markers were selected to provide preliminary insights into the antioxidative and detoxification potential of LAS. The observed upregulation of Nrf-2 in the mycotoxin-challenged birds and its attenuation with LAS supplementation suggest that LAS may support the antioxidant defense system, potentially through activation of the Nrf2-ARE (antioxidant response element) pathway. Similarly, the downregulation of EPHX1 in LAS-treated groups indicates a possible reduction in the need for phase I detoxification of mycotoxins, implying a protective effect at the metabolic level.
We acknowledge, however, that a more detailed understanding of the molecular mechanisms—such as the downstream targets of Nrf-2, involvement of phase II enzymes (e.g., GST, NQO1), and modulation of inflammatory signaling pathways—was beyond the scope of this initial efficacy-focused trial.
Round 2
Reviewer 3 Report
Comments and Suggestions for Authors
Dear Authors,
It is OK.